# Enhancing Therapeutic Efficacy of FLT3 Inhibitors with Combination Therapy for Treatment of Acute Myeloid Leukemia

**DOI:** 10.3390/ijms25179448

**Published:** 2024-08-30

**Authors:** Malia E. Leifheit, Gunnar Johnson, Timothy M. Kuzel, Jeffrey R. Schneider, Edward Barker, Hyun D. Yun, Celalettin Ustun, Josef W. Goldufsky, Kajal Gupta, Amanda L. Marzo

**Affiliations:** 1Department of Internal Medicine, Division of Hematology, and Oncology and Cell Therapy, Rush University Medical Center, Chicago, IL 60612, USA; malia_e_leifheit@rush.edu (M.E.L.); johnsongunnar98@gmail.com (G.J.); timothy.kuzel@nm.org (T.M.K.); celalettin_ustun@rush.edu (C.U.); joe.goldufsky@gmail.com (J.W.G.); 2Department of Microbial Pathogens and Immunity, Rush University Medical Center, Chicago, IL 60612, USA; jeffrey_schneider@rush.edu (J.R.S.); edward_barker@rush.edu (E.B.); 3Hematology, Oncology, Veterans Affairs Long Beach Healthcare System, Long Beach, CA 90822, USA; gusehs80@gmail.com; 4Department of Medicine, Division of Hematology, Oncology, School of Medicine, University of California, Irvine, CA 92617, USA; 5Department of Surgery, Rush University Medical Center, Chicago, IL 60612, USA; khgupta@uchicago.edu

**Keywords:** acute myeloid leukemia (AML), gilteritinib, mTOR inhibitors, rapamycin, NK cells

## Abstract

FMS-like tyrosine kinase 3 (FLT3) mutations are genetic changes found in approximately thirty percent of patients with acute myeloid leukemia (AML). FLT3 mutations in AML represent a challenging clinical scenario characterized by a high rate of relapse, even after allogeneic hematopoietic stem cell transplantation (allo-HSCT). The advent of FLT3 tyrosine kinase inhibitors (TKIs), such as midostaurin and gilteritinib, has shown promise in achieving complete remission. However, a substantial proportion of patients still experience relapse following TKI treatment, necessitating innovative therapeutic strategies. This review critically addresses the current landscape of TKI treatments for FLT3+ AML, with a particular focus on gilteritinib. Gilteritinib, a highly selective FLT3 inhibitor, has demonstrated efficacy in targeting the mutant FLT3 receptor, thereby inhibiting aberrant signaling pathways that drive leukemic proliferation. However, monotherapy with TKIs may not be sufficient to eradicate AML blasts. Specifically, we provide evidence for integrating gilteritinib with mammalian targets of rapamycin (mTOR) inhibitors and interleukin-15 (IL-15) complexes. The combination of gilteritinib, mTOR inhibitors, and IL-15 complexes presents a compelling strategy to enhance the eradication of AML blasts and enhance NK cell killing, offering a potential for improved patient outcomes.

## 1. Introduction

Acute myeloid leukemia (AML) is a heterogeneous malignancy of the myeloid lineage stem cell precursors that give rise to red blood cells, platelets, and white blood cells (not including B and T cells). AML is characterized by chromosomal abnormalities, recurrent gene mutations, and epigenetic modifications that affect chromatin structure [1,2]. AML is one of the most aggressive forms of adult acute leukemia with a high unmet clinical need [3]. The accrued genetic mutations result in the overproduction of neoplastic clonal myeloid stem cells, creating a massive risk for the patient due to decreased ability to clear infections [1]. AML accounts for 1.2% of annual cancer diagnoses in the US and one-third of all leukemia diagnoses and has a 50% relapse rate. In the clinic, patients with AML are diagnosed via routine bloodwork or through the manifestation of symptomatic complications, such as infection, bleeding, or disseminated vascular coagulation. Nonetheless, a bone marrow (BM) biopsy is essential for assessing cytogenic risk and the presence of any genetic mutations or chromosomal abnormalities. Cytogenic risk correlates with patient prognosis and places patients into three risk groups: favorable, intermediate, or adverse risk, where survival rates are 64%, 41%, and 11%, respectively [1,4,5]. 

The mutations in the gene encoding FMS-like tyrosine kinase three, also known as FLT3 mutation, affect one-third of patients with AML and are correlated with poor prognosis [6]. The mutation causes perpetual activation of the FLT3 receptor in the absence of the FLT3 ligand, causing increased proliferation and decreased apoptosis [7,8]. Over the past few years, various FLT3 inhibitors have been developed to target and treat patients harboring the FLT3 mutation. Despite these developments, many patients experience AML relapse and resistance to tyrosine kinase inhibitor (TKI) treatment [9]. Due to persistent activation, the FLT3 mutation also causes constituent signaling of the phosphatidylinositol 3-kinase (PI3K) pathway, leading to dysregulation in the mTOR pathway [10]. Various mTOR inhibitors, such as rapamycin, are effective in treating solid tumors [11]. Because of its involvement in the pathways downstream of FLT3, inhibiting the mTOR pathway is of utmost interest for treating TKI-resistant AML. Combining TKIs with immunotherapies, such as IL-15, CAR-T/NK cells, and TGF-β agonists, aims to mobilize defective and exhausted natural killer (NK) cells against leukemic blasts [12,13,14]. AML is a multifaceted disease, and in this review, we focus on the treatment of FLT3+ AML patients with an emphasis on using FLT3 inhibitors in combination with other therapies. Therefore, approaching it from many angles will optimize and enhance the therapeutic benefits of TKI treatment.

## 2. LSCs and Relapse

AML is a blood malignancy; therefore, it affects every organ system in the body. Leukemic cells possess an innate ability unique to leukocytes for migration and infiltration, unlike solid tumors that require a barrage of cellular changes and reprogramming [15]. Leukemic stem cells (LSCs) constitute a significant source for refractory disease or relapse. LSCs are therapy-resistant cells with persistent self-renewal capacity that drive clonal outgrowth and are essential in the context of extramedullary spread [16,17,18]. As well as being therapy-resistant [16,17,18], they are also marked by the acquisition of pro-survival and pro-proliferative mutations that may emerge as the dominant subclone throughout disease progression [19] and serve as a reservoir for disease relapse after conventional treatments [18,20]. AML is not characterized as a metastatic disease; instead, AML blasts are highly efficient models of “metastatic” spread. Tissues most commonly affected by uncontrolled AML proliferation include the BM, lymph node, liver, spleen, central nervous system, skin, and testicles [15,21]. Disease progression and prognosis depend on the extramedullary spread to these susceptible organ systems, and the interaction of malignant cells with specific tissue microenvironments can create unique heterogeneous responses in patients [15].

Healthy, steady-state hematopoietic stem cells (HSCs) experience exhaustion, quiescence, and dormancy to regulate normal hematopoiesis. The gradual accumulation of mutations in LSCs causes clonal evolution and subsequent dominance over the site of origin. HSCs reside in the hypoxic environment of the BM, and this niche is vital in maintaining and retaining LSCs [22]. LSCs are a highly heterogeneous population with various mutations contributing to their unregulated BM growth [23]. The BM niche, regulated by crosstalk with multiple cell types, maintains stem cells and serves as a reservoir for hematopoiesis. However, LSCs rely on this crosstalk within the BM niche to maintain their survival and proliferative properties [22,23,24]. A variety of signaling pathways, including SDF1α/CXCR4, Wnt/β-catenin, VCAM/VLA-4/NFκB, and PI3K/Akt/mTOR pathways, positively support the development of LSCs and contribute to drug and chemotherapy resistance [25]. AML relapse occurs due to therapy-resistant minimal residual disease (MRD), in which LSCs remain in the BM environment post-treatment [22,26]. These remaining LSCs contribute to relapse by maintaining their gene expression and properties, allowing stemness and self-renewal [27]. Therefore, identifying and targeting the LSC compartment is of utmost importance to prevent AML relapse following treatment.

## 3. Current Treatments

Current induction therapy for AML patients aims to achieve complete disease response (CR) with no measurable residual disease (MRD) [1,28]. Induction treatment can be combined with additional targeted therapy to enhance effectiveness. Using patient risk groups, those who are in favorable or adverse risk groups and are medically fit enough to undergo chemotherapy will receive a continuous infusion of cytarabine for seven days, along with short infusions of anthracycline for the first three days, dubbing the treatment 7 + 3. Typically, 7 + 3 therapy causes CR in up to 90% of patients in the favorable risk group and 50–70% CR in an intermediate or high-risk group [29]. Between 50% and 80% of patients in the adverse risk group may relapse without allo-HCT due to treatment resistance [1,29]. However, many AML patients will experience relapse due to LSC survival post-treatment, making AML treatment very challenging. Two of the most commonly used treatments for AML are induction therapy, consisting of cytotoxic chemotherapy and hypomethylating agents, followed by post-remission or remission consolidation therapy, which can include continuation treatment with chemotherapy, like cytarabine or allo-HCT transplant, depending on favorable vs. unfavorable risk assessment, respectively [1].

Allo-HCT is commonly used in those at high risk of relapse following their first CR [30]; however, it is associated with a morbidity and moderate non-relapse mortality (NRM) rate of 20% [31]. Relapse rates following first allo-HSCT are 20–40% depending upon relapse risk of AML. Allo-HCT aims to replace diseased BM with a healthy BM while inducing a graft-versus-leukemia (GVL) response against leukemic blasts from donor immune cells. According to the intensity of chemo/radiation therapy before allo-HCT (i.e., conditioning or preparative regimen), there are two main regimens: HCT myeloablative regimen (MAC) and non-myeloablative (NMA) conditioning. NMA conditioning regimens can be further divided into two subgroups by the intensity of the regimens: moderate intensity [i.e., reduced-intensity conditioning (RIC)] or very mild intensity (NMA) [32]. It has gained popularity because of its tolerability, particularly among older or frail patients [33]. However, relapse is more common with NMA/RIC [34,35,36,37,38]. To understand the alterations in a patient’s immune system following allo-HCT, it is essential to investigate the pre- and post-treatment effects [32]. Notably, even healthy donor myeloid cells may display deficiencies in their maturation and activation processes in the recipient BM. These defects can potentially lead to disease relapse, though this form of relapse compromises a small percentage of patients [39]. Given the notably high relapse rates observed in patients with AML following intensive therapy regimens, the exploration of novel treatment approaches is of paramount significance to enhance patient survival and minimize the risk of disease recurrence [40]. Maintenance therapies after allo-HCT with a targeted small-molecule inhibitor (e.g., FLT3 inhibitors) have been investigated to decrease relapse.

## 4. Genetic Mutations in AML

There is considerable genetic diversity across AML, but despite this heterogeneity, many genetic alterations are prevalent and used in assessing patient cytogenic risk [41]. However, AML disease genetics can be quite complex, making risk assessment and treatment decisions difficult for clinicians. Recent advances in sequencing technology, such as karyotyping, chromosomal microarray analysis (CMA), and next-generation sequencing (NGS), have revealed multiple newer mutations that play a role in patient diagnosis and prognosis [41,42]. Six examples of common mutations in AML include FMS-like tyrosine kinase 3 (FLT3), nucleophosmin 1 (NPM1), CCAAT/enhancer binding protein alpha (CEBPA), runt-related transcription factor 1 (RUNX1), additional sex combs-like 1 (ASXL1), and tumor protein p53 (TP53) [41,43]. 

## 5. FLT3 Mutation

One of the most common recurring mutations in AML is the FLT3 mutation, which is present in 30% of all AML cases [44]. Patients with the FLT3 mutation tend to be younger and generally respond well to initial chemotherapy but have a much higher relapse rate, decreased survival rates, poorer response to the subsequent treatment, and shorter relapse-free survival (RFS) when compared to patients who lack the mutation [7,45]. 

The FLT3 mutation has two forms: the FLT3 internal tandem duplication (FLT3-ITD) mutation and the FLT3 tyrosine kinase domain (FLT3-TKD) mutation. FLT3-ITD accounts for 25% of AML cases, while FLT3-TKD is seen in 5% of cases [46,47]. As a result of these mutations, the FLT3 receptor is perpetually activated in the absence of the FLT3 ligand (FLT3L), increasing proliferation and decreasing apoptosis [46,47], making FLT3 an excellent target for cancer therapies. Various FLT3 inhibitors have been developed to block FLT3 activation. Different first-generation and next-generation inhibitors exist, but the FDA has approved midostaurin, gilteritinib, and, more recently, quizartinib for use in the clinic [7,48]. First-generation inhibitors are less specific than next-generation FLT3 inhibitors and have more off-target effects than next-generation inhibitors. FLT3 inhibitors are further subdivided into type I and type II inhibitors based on their ability to target either form of the mutation (FLT3-TKD or FLT3-ITD). Type I inhibitors bind and inhibit both the active and inactive states of the mutated receptor, while type II inhibitors restrictively bind to the active form of the receptor [8,48]. Some type II inhibitors do not entirely block the FLT3 mutation, which causes secondary drug resistance, most likely caused by a single amino acid substitution within the D835 codon [48]. In contrast, type I inhibitors have increased sensitivity to both forms of the FLT3 mutation, causing less secondary drug resistance [8,48]. 

FLT3 pathway: The downstream targets of the FLT3 pathway include PI3K/AKT, JAK2/STAT5, and MAPK. When bound to the FLT3 ligand (FLT3-L), trans-autophosphorylation of tyrosine residues in FLT3 results in binding of adaptor proteins (e.g., SHP2, GRB2, and SRC family kinases), mainly leading to activation of the PI3K/Akt/mTOR and RAS/MEK/ERK pathways, but FLT3-ITD also induces JAK/STAT signaling through phosphorylation of STAT5A (Figure 1) [10,48,49]. On the other hand, FLT3-TKD shows increased activation of SHP1 and SHP2, where SHP1 negatively regulates the JAK signaling pathway, leading to low levels of JAK2 and STAT3 activity [8]. 

After PI3K recruits scaffold proteins, Akt is partially phosphorylated via PDK1 at Thr308. Subsequent phosphorylation at Ser473 by the mTOR 2 complex (mTORC2) and members of the PI3K-related kinase (PIKK) family fully activates Akt [45]. The PI3K/Akt/mTOR pathway has been extensively studied in both normal and malignant cells. In AML, the PI3K/Akt/mTOR pathway is upregulated due to metabolic reprogramming to a constitutively active state and is present in around 60% of AML cases [45,50,51,52,53,54]. This pathway is central to hematopoietic stem cells and regulates their essential functions, such as proliferation, differentiation, and survival. 

## 6. Targeting FLT3

Since both ITD and TKD mutations cause the FLT3 receptor to be constitutively active, this makes the FLT3 receptor an excellent target for inhibition. Nearly a dozen FLT3 inhibitors and many novel inhibitors are currently being tested in clinical trials [55,56]. Tyrosine kinase FLT3 inhibitors can be divided into two groups based on their specific ability to inhibit FLT3 and associated downstream pathways [8,55,57,58]. 

### 6.1. First-Generation FLT3 Inhibitors

First-generation inhibitors are TKI with a broad anti-kinase activity repurposed for use in FLT3-mutated (mFLT3) AML patients [56]. Both first- and second-generation inhibitors can be broken into type I and type II inhibitors [35,36,44,45,46,48]. Type I inhibitors, such as midostaurin and lestaurtinib, bind to the ATP-binding site in the intracellular active pocket of the enzyme and are active against both the ITD and TKD forms of the mutation. Meanwhile, type II inhibitors sunitinib and sorafenib bind to the ATP-binding site and interact with the adjacent hydrophobic pocket, which is exposed in the inactive conformation, thus rendering type II inhibitors ineffective against the TKD mutation [56]. 

In vitro, first-generation inhibitors appeared to be potent against mFLT3, but when tested in vivo, the results were highly disappointing, with low efficacy and high adverse effects [57,59,60,61,62,63]. Their broad anti-kinase activity causes higher toxicity profiles when compared to second-generation TKI. However, first-generation TKI has been successful when combined with chemotherapy during induction therapy [56]. 

### 6.2. Midostaurin

Midostaurin is an active type I inhibitor against wild-type FLT3 and mFLT3 mutations, KIT, PDGFR, VEGFR2, and other members of the protein kinase C family [56,64]. According to the phase III RATIFY trial, midostaurin significantly improves overall and event-free survival of patients with mFLT3 AML in combination with intensive 7 + 3 chemotherapy compared to a placebo but saw more than half of the patients who had achieved CR after midostaurin treatment relapse [4,57,65]. Tested in combination with hypomethylating agents (HMAs), midostaurin exhibited limited activity and was not well tolerated due to hematologic toxicity and high rates of infection [55,66,67]. Midostaurin used for post-allo-HSCT treatment is well tolerated, with patients achieving a median OS of 26 months [68]. In the phase II RADIUS trial, 60 randomized patients who underwent allo-HSCT in CR1, where most did not receive midostaurin during induction, received midostaurin for one-year post-HSCT. Still, the study could not detect a difference in relapse-free survival (RFS) [69]. However, midostaurin has since been used routinely to induce AML patients with FLT3 mutations who are fit enough to receive chemotherapy (Table 1) [55].

### 6.3. Sorafenib

Sorafenib is an oral multi-kinase inhibitor of RAF-1, VEGF, c-KIT, PDGFR, Erk, and FLT3 approved for treating hepatocellular and renal cell carcinomas and off-label use in AML post-HSCT therapy [74,75]. Sorafenib has demonstrated the ability to suppress FLT3-ITD activity in AML patients effectively. Furthermore, it exhibits modest therapeutic efficacy as a single agent when employed in cases of relapsed or refractory AML [75]. It is hypothesized that after allo-HSCT, there is a synergy between sorafenib and allogeneic immunity [76]. In the phase II SORMAIN trial, the 24-month RFS probability of individuals treated with sorafenib was 85% compared to 53.3% with a placebo [77]. Sorafenib has been shown to be effective in combination with chemotherapy when used as part of induction therapy. The phase II SORAML trial tested sorafenib combined with intensive chemotherapy followed by sorafenib for maintenance versus chemotherapy alone and found that the participants on sorafenib had a significantly longer event-free survival than the group who received chemotherapy alone. Still, the overall survival (OS) between groups was insignificant (Table 1) [78].

### 6.4. Second-Generation FLT3 Inhibitors

The development of second-generation inhibitors relied heavily on the inadequacies of their first-generation counterparts. Drugs such as gilteritinib, quizartinib, and crenolanib were designed to specifically target FLT3, thus rendering them more selective and potent and lessening off-target toxicity. This is reflected in their ability to trigger myeloid differentiation and exhibit more significant clinical activity as monotherapy [8,56,57]. Gilteritinib and crenolanib are type I inhibitors that are active against the active and inactive conformations of the FLT3 receptor. In contrast, quizartinib, a type II inhibitor, is only active against the inactive form of mFLT3 (Table 1) [47].

### 6.5. Quizartinib

Quizartinib is a highly selective and potent second-generation FLT3 inhibitor dosed at 1 mg/kg once a day. In a phase I dose escalation trial, quizartinib was given in combination with chemotherapy in 19 patients with newly diagnosed AML. Of the 16 patients who achieved a good response, 14 achieved CR, and 2 reached a morphologic leukemia-free state with low-grade toxicity [74,79]. Another dose escalation trial administered quizartinib as maintenance therapy to 13 patients post-allo-HSCT. Two patients receiving 40 and 60 mg per day terminated their treatment early due to grade 3 gastric bleeding and anemia; however, no maximum tolerated dose was reported, and 60 mg per day was the highest dose they studied [80]. It has also been demonstrated that quizartinib is highly active against relapsed and refractory AML. In a phase I trial by Cortes et al., out of 76 patients, 23 responded to treatment, where 10 achieved CR and 13 achieved partial remission [81]. Adverse effects that arise, such as cardiac dysfunction, raise concerns about toxicity even at low concentrations [82,83]. In July 2023, the FDA approved quizartinib for mFLT3 AML patients. This decision was based on the phase 3 QuANTUM-First study that evaluated its effectiveness and safety in a placebo-controlled study of 539 FLT3+ AML patients [84]. Patients received either VANFLYTA (*n* = 268) or a placebo (*n* = 271) in combination with induction and consolidation therapy and as maintenance monotherapy [85]. The CR rate in patients receiving VANFLYTA was 55%, with a median duration of CR of 38.6 months, while the placebo had a CR rate of 55% and a median duration of CR of 12.4 months [84,85]. Patients survived twice as long when quizartinib was added to standard treatment compared to the placebo group. This prompted the FDA to approve VANFLYTA in July 2023 as a generally well-tolerated treatment for mFLT3 AML patients. 

### 6.6. Gilteritinib

Gilteritinib is a second-generation type I inhibitor approved for use in patients with relapsed or refractory leukemia. One feature of gilteritinib is that it can mobilize NK cells and improve NK activation and function since the drug enhances cytokine production by myelomas [7,48,86]. Another essential feature of gilteritinib is its ability to target active and inactive forms of the FLT3-TKD and FLT3-ITD mutations [9]. The ADMIRAL trial demonstrated that patients treated with 80 mg/d of gilteritinib have a tumor objective response rate (ORR) of 40% of all AML patients, regardless of whether they displayed the mutation—an ORR of 52% in those with the FLT3 mutation [87]. The findings from this trial solidified gilteritinib as a treatment for relapsed/refractory AML due to its significantly longer survival rates compared to those who only received chemotherapy. It is of note that 40% of patients (16/40) who received allo-HCT following gilteritinib maintenance survived for two years relapse-free, whereas 8% of patients without gilteritinib maintenance post-allo-HCT survived [88,89]. Gilteritinib also has less off-target effects due to its ability to inhibit the tyrosine kinase AXL, a membrane-bound receptor kinase associated with FLT3 that activates the same pathways as FLT3 [7]. Patients receiving gilteritinib also experienced less severe adverse events while taking the drug, demonstrating a favorable safety and toxicity profile. Gilteritinib is one of the most promising FLT3 inhibitors, and ongoing trials are testing the drug in combination with venetoclax and hypomethylating agents for relapsed and refractory AML [90,91]. The FDA approved gilteritinib for use in November 2018 in relapsed/refractory AML [92]. Table 2 illustrates the FLT3 inhibitors that are in clinical trial.

## 7. TKI Resistance and Disease Relapse

The activation of FLT3 leads to downstream activation of intracellular signaling pathways, including RAS/MAPK, Pi3K/AKT/mTOR, and JAK/STAT5. Clonal selection during TKI treatment can select for gain-of-function gene mutations in these pro-survival pathways, ultimately leading to treatment resistance [83]. Eradication of mutated resistant clones is of utmost importance to avoid and treat relapsed/refractory AML. It was reported that 55% of patients who had initially responded to treatment and experienced a relapse had treatment-emergent mutations at the time of relapse, including on-target mutations in *FLT3* and a variety of off-target epigenetic modifiers such as the *RAS/MAPK* genes, where 13% of patients developed resistance TKI treatment [93]. Patients treated with type II FLT3 inhibitors that have developed RAS/MAPK mutations are associated with inferior survival, and 29% of those treated with type I FLT3 inhibitors developed RAS/MAPK resistance at relapse [93]. Improving our knowledge of secondary resistance patterns to FLT3 inhibitors and strategic use of classical chemotherapy can delay the onset of resistance and relapse by combining FLT3 inhibitors with chemotherapy [46].

The incidence of disease relapse can increase in the 30 days immediately following allo-HCT, especially in patients with measurable residual disease (MRD)-positive and FLT3-positive AML [77]. One of the main concerns following allo-HCT is thrombocytopenia. However, gilteritinib has shown to be less active than the other FLT3 inhibitors in suppressing the necessary signals for hematopoiesis, allowing gilteritinib maintenance treatment to begin even if the patient has a low platelet count [7]. Because of this, patients with thrombocytopenia can begin maintenance treatment earlier, which may contribute to the suppression of AML relapse in those 30 days post-allo-HCT [88]. Besides direct action on AML cells, indirect effects of FLT3 inhibitors, such as weakened expression of PD-1 and TIGIT within donor CD8+ T cells and increased production of IL-15, enhance the graft-versus-leukemia effect. Therefore, especially in the case of patients with MRD+FLT3+ AML, it may be of use to continue maintenance treatment past the 2-year mark, as FLT3 inhibitors can act on residual leukemic cells to produce IL-15 and maintain GVL response [88,94,95]. IL-15 would also have an additional effect by enhancing NK cells’ response to leukemia cells through increased activation receptors and enhanced cytolytic potential [96].

## 8. The mTOR Pathway

The mammalian target of the rapamycin (mTOR) pathway exists as several complexes (mTORC1, mTORC2, mTORC3, and mTORC4) (Figure 2) that control proliferation, survival, autophagy, metabolism, and immunity [11,97,98,99]. Because of this, the mTOR pathway is of utmost interest in cancer, where these pathways become disrupted and dysregulated, causing unchecked cell proliferation and a lack of apoptotic activity. The mTOR complexes are all very similar in structure, where they all share mTOR as part of their central structure but differ mainly in the activating proteins of each complex. mTORC1, mTORC2, and mTORC4 contain mammalian lethal with Sec13 protein 8, also known as GβL, and mLST8 as part of their structure, but otherwise, these complexes vary in their catalytic subunits. mTORC1 also includes regulatory-associated protein of mTOR (Raptor), PRAS40, and DEPTOR, all of which are specific to the mTORC1 complex. Meanwhile, Rictor and MAPKAP1 are unique to the structure of mTORC2 [100,101]. For now, mTORC3 contains ETV7 and other undefined components [99], while mTORC4 includes mEAK7 and DNA-PKcs, which are hypothesized to play roles in tumorigenesis and anti-cancer drug resistance by controlling T cell’s response to DNA damage and cancer cell physiology [97,102,103]. Although there is minimal information on the functional role mTORC3 and mTORC4 play, it is anticipated they both play a role in tumor immune response and treatment evasion. It is also likely that other mTOR complexes will emerge [19].

The mTORC1 complex is the most well-studied of the complexes in the mTOR pathway. Its activity is regulated by growth factors, cellular energy, stress, and other factors like cytokines and self-amino acids that work to activate and inhibit the complex [104]. Activation of mTORC1 by growth factors such as the Ras homolog (RHEB-dependent), a GTPase that directly interacts with mTOR, results in its activation being enriched in the brain (RHEB-dependent) [104]. When bound to epidermal or insulin-like growth factors (EGF and IGF), the receptors are activated, turning the PI3K–PDK1–Akt signaling pathway on, where Akt then activates tuberous sclerosis complex 2 (TSC2), inhibiting the TSC complex [105,106]. The TSC complex can then inactivate RHEB, thereby inhibiting mTOR and activating mTORC1 [11,107]. However, the rapamycin-resistant mTORC2 can activate the IGF–IR–Akt axis, which can upregulate the activity of mTORC1 [108]. mTORC2 is less well studied than mTORC1, but more studies regarding mTORC2 may indicate that it is the leading promoter of cell proliferation and survival in AML. Not much is known about the activity of mTORC3 and mTORC4, but both exhibit resistance to rapamycin. 

Inhibition of the mTOR pathway is of utmost interest for treating mFLT3 AML patients due to constitutively active FLT3. One such method of inhibition is achieved via rapamycin and its analogs (rapalogs), which is a potent inhibitor of mTORC1. Rapamycin was first discovered as an antifungal, immunosuppressive, and antiproliferative agent, but since it is not water-soluble, it is unsuitable for treating human cancers [109,110]. Thus, rapalogs such as everolimus and temsirolimus have been used in the clinic, specifically for treating renal cell carcinoma [11]. It is of note that rapamycin alone is ineffective in treating solid tumors and is best used in combination with other therapeutic agents. Also, inhibition of only mTORC1 can cause feedback activation of IGF-IR and Akt, compromising rapamycin’s anti-cancer effects [111]. Thus, targeting of upstream pathways (e.g., inhibiting FLT3 via gilteritinib) in addition to treatment with rapamycin may show less resistance, better anti-tumor effects (Figure 3), and better mobilization of NK cells [111]. 

Inhibition of the remaining three mTOR complexes cannot be achieved via rapamycin or its analogs as these complexes are resistant to said rapalogs. Therefore, selective inhibitors and ATP-competitive inhibitors have been developed to block the mTORC2 rapamycin-resistant complex. Small-molecule inhibitors of the mTORC2 kinase (torkinibs), such as JR-AB2-011, are designed to arrest cell development and induce apoptosis by targeting mTORC2 [112]. Additionally, it has been shown that inhibiting mTORC2 in conjunction with mTORC1 can reverse rapamycin resistance in the mTORC1 pathway by limiting its feedback activation [11]. However, preliminary data from our laboratory have shown that treating an AML FLT3+ cell line, MOLM-14, in vitro with mTORC1 and mTORC2 does not induce cell death but significantly inhibits cell proliferation (unpublished data). Therefore, adding another agent, such as the FLT3 inhibitor gilteritinib, could improve the treatment of FLT3-resistant AML patients. No inhibitors are known to block mTORC3 and mTORC4, but further research may reveal inhibitors for these complexes.

**Figure 3 ijms-25-09448-f003:**
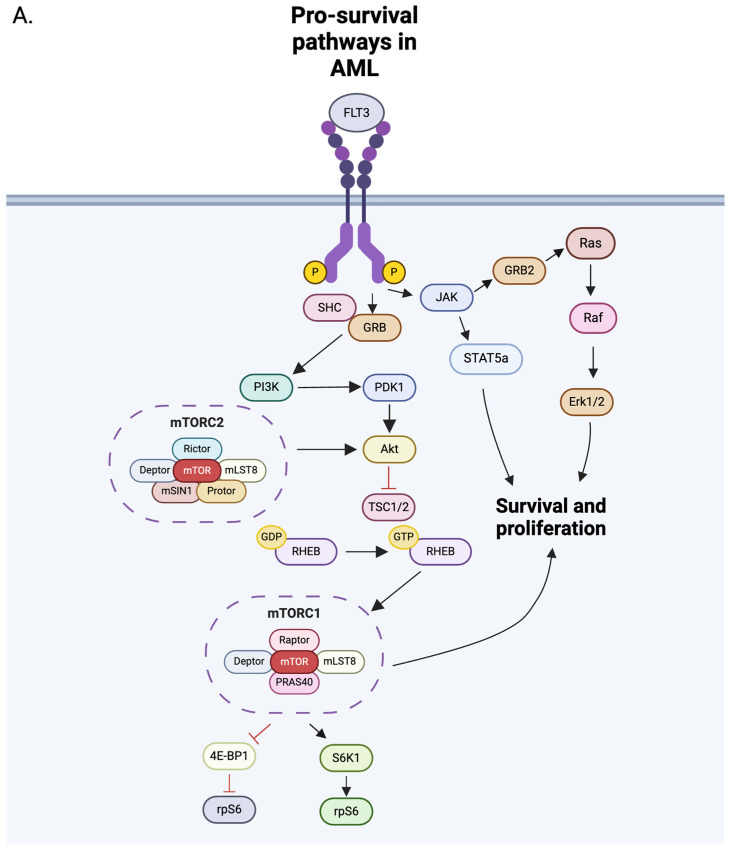
(**A**) Pro-survival signaling pathways in AML and (**B**) proposed inhibition mechanisms using FLT3 and mTOR inhibitors. FLT3 mutations, such as ITD and TKD, lead to dysregulation of the FLT3 pathway, causing constitutive activation. Gilteritinib inhibits FLT3 at the ITD in the juxtamembrane domain and the TKD mutation in TKD2. Small-molecule inhibitors of the mTOR complexes include dual PI3K/mTORi, allosteric mTORi, and ATP competitive inhibitors of mTORC1 and mTORC2. The red arrow denotes elevated Akt phosphorylation, and the blunt-end red lines represent negative regulation. Black blunted ends represent mechanisms of therapeutic intervention [113,114]. These figures were generated using Biorender.

## 9. NK Cells

NK cells are innate immune cells that display antileukemic effects mediated by two mechanisms: (1) destruction of the leukemic cell through perforin/granzymes or (2) through engagement of death receptors via FAS ligand or TRAIL. While these mechanisms of NK cell destruction of tumor cells are similar to what is observed with CD8+ T-cell-mediated lysis of tumor cells, NK cells mediate this response independent of MHC-class-I-based antigen recognition. In contrast, NK cells continuously circulate in the blood, lymph nodes, and tissues to rapidly identify and eliminate stressed, infected, or cancer cells [115]. NK cells are regulated through a delicate balance of activating (e.g., NCR, NKG2D, and DNAM-1) and inhibitory (e.g., NKKG2A and inhibitory KIRs) receptors [116]. Besides the expression of cytolytic molecules, NK cells can express inflammatory cytokines in response to proinflammatory cytokines or when their activation receptors engage ligands on cancer cells. Thus, NK cells have the capacity to trigger and activate the adaptive immune system, making them one of the most critical cells in defense against cancers, specifically AML, due to their unique proinflammatory response to cancer [117].

While inhibitory receptors, such as NKG2A, on NK cells can recognize MHC class I molecules (Figure 4), there are instances where NK cells can be activated by MHC class I molecules. Engagement of NKG2C/CD94 with HLA-E causes NK cell activation using a mechanism similar to the TCR interaction with MHC (Figure 4). Activation of NK cells leads to NK effector functions such as degranulation, which is indicated by the expression of cluster of differentiation 107a (CD107a), also known as lysosomal-associated membrane protein 1 (LAMP1), expression on the cell surface. CD107a is primarily expressed within the inner leaflet of the lysosomal membrane but is upregulated on the membrane surface of cytotoxic T and NK cells’ plasma [118,119]. HLA-E is found to be upregulated on tumor cells due to IFN-γ response genes located upstream on the HLA-E gene when immune effector cells reach the site of infection and secrete IFN-γ [120]. The interaction of NKG2C/CD94 and HLA-E is also crucial for further activating allo-NK cells and some cytotoxic CD8+ T cells. 

In AML, NK cells are severely impaired, similar to other lineages that show hypo-responsiveness and exhaustion in cancer [121]. AML cells have also developed various methods of evading the immune system and its antileukemic effector cells, including the increased ability to activate regulatory T (Treg) cells [122]. These Treg cells inhibit NK cell activation and limit their polyfunctionality, thus tightly regulating host self-destruction [117]. NK cells in AML also exhibit increased TIM-3 expression, a significant checkpoint inhibitor of NK cell function and an indicator of NK exhaustion and restriction following long-term infections [123,124]. Another method of AML cell immune evasion that directly hinders NK cell function is to downmodulate important NK-cell-activating ligands, such as HLA-E [111], or shed soluble ligands, such as MICA/B to the NKG2D activation receptor, which collectively lower engagement of NK cell activation receptors [104,105,112,125]. 

Developing strategies to enhance NK cell function after allo-HCT, such as in vivo expansion using cytokines, have been promising advancements in helping patients achieve CR. It was previously found that adoptively transferred NK cells from haploidentical donors can be expanded in vivo with high doses of immunosuppressants and exogenous IL-2; however, CR was observed in 5 out of the 19 patients in the trial [126]. However, treatment with IL-2 drives the expansion of immunosuppressive Tregs due to their increased expression of the high-affinity IL-2 receptor α chain, CD25, which is indispensable for mediating Treg survival and suppressor functions [127]. Interestingly, it was discovered that NK cell adoptive therapy causes a noticeable increase in endogenous IL-15 levels, which produces potent proliferative signals that expand NK cells [126]. Treatment with intravenous (IV) or subcutaneous (SC) administration of recombinant human IL-15 (rhIL-15) is associated with an antileukemic effect when combined with lymphodepleting chemotherapy and haplo-NK-cell infusion and caused CR in 35% of patients, demonstrating the importance of NK cells in harnessing an immune response against AML [128]. Thus, using IL-15 for in vivo expansion of adoptively transferred NK cells is a promising treatment strategy, but further strategies to prevent host rejection must be explored.

## 10. Engaging Immune Cells with IL-15

The γc-cytokine interleukin-15 (IL-15) has various functions in the development and progression of hematological malignancies but also has a crucial role in the cytotoxic boost against cancer cells by serving as a growth factor for both T and NK cells [12]. IL-15 has shown anti-cancer effects in many preclinical models [116,117,118,119,120,129,130,131,132,133], and presently, many clinical trials are testing its efficacy in cancer immunotherapy [134,135,136,137,138,139,140]. However, the potential of using IL-15 to mobilize substantial effector populations against AML and induce graft-versus-leukemia responses is hindered due to toxicity associated with IL-15 infusion. This has led some studies to turn to intratumorally injecting IL-15 [141]. Still, since AML is located systemically in the circulatory system, IL-15 must be given as an infusion, posing a higher toxicity risk to the patient [12]. Molecules such as IL-15 agonists with modified structures that mimic physiologic IL-15 are being explored to avoid potential adverse events. Various ongoing clinical trials are investigating the effectiveness of these small molecules. Still, most have found that IL-15 is not enough as a single agent and are evaluating the efficacy of IL-15 combination therapy with immune checkpoint blockades, monoclonal antibodies, and bispecific antibodies [12]. 

The most recent literature on IL-15 immunotherapy exists in adoptive cell therapies that expand CAR-T and CAR-NK in combination with cellular infusion to maintain cell therapies in vivo. One study developed anti-NKG2C/IL-15/anti-CD33 killer engager cells that triggered cell-mediated cytotoxicity against primary AML blasts [142]. Similarly, IL-15/IL-15Ra/CD80-expressing AML cells have been used as a post-remission vaccine and reported 50% OS in in vivo mouse models of AML [143]. However, adding systemic IL-15 with CAR-NK cell therapy has been shown to exhibit lower clinical activity than IL-2 [144]. A novel tri-specific killer cell engager (TriKE) was developed to overcome the limitations of harnessing NK cells for cancer immunotherapy [132]. These TriKEs were designed by integrating IL-15 into the current BiKE to promote NK cell activation, expansion, and survival. The TriKEs also crosslinked tumors and NK cells and restored NK cell function in patients with myelodysplastic syndrome (MDS) [13,14]. The purpose of TriKEs was to traffic NK cells to the tumor, trigger ADCC when bound to CD16, and drive in vivo expansion [145]. Compared to BiKEs, TriKEs elicit a more significant NK cell effector response to CD33+ myeloma cell lines and primary AML blasts. They were also superior to BiKEs in restoring NK cell function in samples from patients undergoing allo-HCT [13,145,146]. Based on good outcomes and an increased safety profile, TriKEs are a promising treatment strategy for AML. 

## 11. Engaging Immune Cells by Blocking TGF-β

Transforming growth factor β (TGF-β) is a secreted cytokine with a myriad of physiological and pathological processes affecting many subsets of immune cells [147,148]. TGF-β impairs the function of MHC class II, causing impaired priming of CD4 T cells as well as suppression of NK cell and CD8 T cell activity by inhibiting differentiation, proliferation, and effector functions [147,148,149,150,151]. Meanwhile, TGF-β also promotes the differentiation of suppressive immune cell subsets [152,153,154]. Normal physiological functions of TGF-β show it has a crucial role in immune suppression, preventing chronic inflammation and tolerance, especially in the gastrointestinal tract [149,155,156]. Still, in malignant disease, TGF-β promotes immune escape, tumor progression, and extramedullary spread [149,157,158,159,160] through various mechanisms, such as release of soluble NKG2Ls via proteolytic cleavage or exosomal release of membrane-bound NKG2DL [161,162,163,164,165].

Interestingly, much evidence is emerging that TGF-β impairs NK and cytotoxic T cell immune recognition of tumors by downregulating the activating receptor NKG2D, hindering tumor removal by these cytotoxic lymphocytes [147,166]. Therefore, targeting TGF-β is an attractive method of boosting tumor immunity and has a possible role in cancer treatment. It is hypothesized that rescuing the NKG2D–NKG2DL axis via blocking TGF-β will potentially increase tumor elimination by utilizing NKG2D-mediated tumor recognition, and various cancer therapies targeting the TGF-β are being investigated or are in clinical trials. Novel drugs, such as galunisertib, sotatercept, luspatercept, and vactosertib, have produced remarkable results in phase I/II studies. For example, galunistertib functions by inhibiting the TGF-βRI kinase, preventing downregulation of surface NKG2D by TGF-β and also enhancing the anti-tumor effect of adoptively transferred cytotoxic lymphocytes with low side effects [167,168,169]. Targeted delivery of cytokines has also been shown to improve NK and T cell effector functions. A combination of IL-2 and IL-18 shielded NK-92MI cells from NKG2D downregulation [159], and an IL-15 super agonist/IL-15R*α* rescued tumor cells from immunosuppression [170,171]. Transduction of NK cells with a dominant negative form of TGF-βRII before adoptive transfer has been shown to shield NKG2D from the suppressive effects of TGF-β and supports maintenance of surface NKG2D [172,173]. A small cohort of chemorefractory Hodgkin lymphoma shows that four out of seven patients achieve CR after treatment with transduced cells, suggesting the safety and efficacy of this type of treatment [174]. Another strategy to convert TGF-β signals into stimulatory ones uses CAR-T cells [175], but no in vivo studies have been conducted in solid or hematological tumors. 

The importance of another protein, SMAD4, in the positive regulation of NK cells has been elucidated by Wang et al. 2018, where SMAD4 acts as a mediator of TGF-β signaling pathways [176] and, in cancer, is both a tumor promoter and a tumor suppressor [177,178,179]. In the early stages of cancer, SMAD4 contributes to anti-tumor immunosurveillance by NK cells. Still, it switches in later stages to have a more immunosuppressive role caused by TGF-β produced by the tumor [180]. They also found that deleting the *Smad4* gene impairs NK cells’ maturation, homeostasis, and immune surveillance [180]. Thus, SMAD4 may be a critical protein in regulating NK cells switching from immunosurveillance to immunosuppression. Recently, a study used the tyrosine kinase inhibitor imatinib in combination with TGF-β agonists in patients with imatinib-resistant CML and found that the combination blocked SMAD4 phosphorylation and achieved better growth inhibitory and apoptotic responses than using imatinib alone [181]. Such combinations using TKI with TGF-β agonists in AML have not yet been explored.

## 12. Combination Treatments

Combining AML treatments is an attractive strategy for eliminating relapsed and refractory leukemia. Specifically, combining traditional chemotherapy methods with kinase inhibitors has been an area of interest since kinase activity plays a pivotal role in leukemogenesis. For example, one such combination therapy is the administration of gilteritinib (where gilteritinib was given at a dose of 120 mg daily on days 4 to 17 or 8 to 21 of 7 + 3 induction from day 1) in combination with conventional 7 + 3 induction therapy with daunorubicin or idarubicin and high-dose cytarabine [182]. In this study, patients who harbored the FLT3 mutation (36 out of 58) achieved a CR of 83% after one induction cycle and an OS of 46.1 months, thus demonstrating gilteritinib’s efficacy as an addition to induction for patients with newly diagnosed AML [182]. Similarly, high CR rates were observed when gilteritinib was combined with venetoclax, a BCL2 inhibitor and apoptosis inducer [183,184], regardless of prior FLT3 inhibitor exposure. Thus, applying small-molecule inhibitors in combination with the FLT3 inhibitor gilteritinib is demonstrated to avoid the clonal evolution of leukemic blasts [185]. 

The use of FLT3 inhibitors in post-allo-HSCT maintenance has been shown to improve survival rates and prevent relapse in FLT3+ AML. It has been reported that sorafenib, quizartinib, gilteritinib, and midostaurin are currently being used as maintenance strategies in clinics around the globe [186]. Additionally, a meta-analysis reported the safety and efficacy of FLT3 inhibitor use, where FLT3 inhibitor use significantly improved OS and RFS, and patients on FLT3 inhibitors exhibited a lower CR than patients receiving the placebo [72]. Meanwhile, Fleischmann et al. reported that using the first-generation TKI, sorafenib, as maintenance post-chemotherapy induction still resulted in poor outcomes for patients, in which only two patients achieved a BM response, thus prompting calls for the use of second-generation TKIs to help improve outcomes [187,188].

Combinations of immunotherapy and FLT3 inhibitors have also been shown to enhance drug effectiveness in relapsed and refractory AML. Existing preclinical evidence has shown synergism between sorafenib and allo-immunity to enhance graft-versus-leukemia effects. FLT3 inhibition with sorafenib reduced ATF4 expression, allowing IRF7 activation to cause IL-15 transcription. IL-15 levels were able to engage CD8+ CD107a+ IFNγ+ donor T cells to eliminate AML tumors [90,95]. To explore the graft-versus-leukemia effect in a selective TKI, a mouse model was used to examine IL-15 production and the effects on T cell activation with or without allo-HSCT. Upon FLT3 inhibition, IL-15 production was upregulated, and T cell exhaustion marker expression was reduced. This study demonstrated GVL effects without exacerbating GVHD. However, short-term gilteritinib treatment could not suppress leukemic growth without the involvement of T cells [90,94].

## 13. Conclusions

In many cancers such as breast, pancreatic, liver, lymphoma, and AML, there is a continual problem of the development of tumor resistance, and there is little doubt that these dysregulated pathways are of significant interest to cancer’s ability to grow and wreak havoc on the host’s body. Many current drugs show promise in combatting AML, but few engage the host’s immune system. However, FLT3 inhibitors such as gilteritinib have led to better outcomes for AML patients by reducing patient risk and mortality, enhancing proinflammatory cytokine production by NK cells, and the possibility of combination therapies to combat tumor resistance to treatment better than previously available treatments. Other preclinical evidence has also suggested the role of IL-15 expression post-TKI treatment and its ability to improve immune cell effector functions. While current findings are promising, more studies are needed to validate the synergistic effects of these proposed treatments in the context of relapsed and refractory AML. This research is critical for advancing our understanding of cancer resistance mechanisms and refining our strategies for more effective and durable cancer treatment.

## Figures and Tables

**Figure 1 ijms-25-09448-f001:**
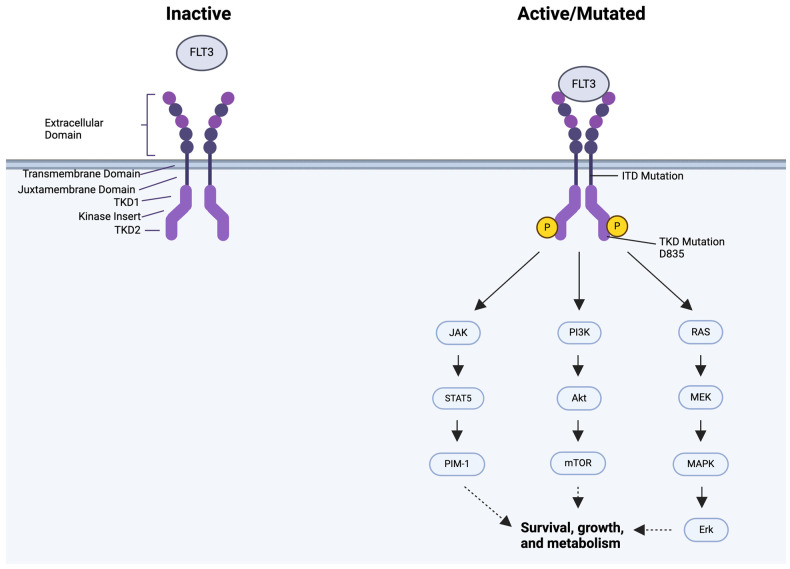
Structure of the FLT3 receptor in the inactive and active/mutated forms and associated downstream. FLT3 ligand (FLT3L) binding to the FLT3 ligand causes receptor dimerization and activation of downstream signaling pathways. Mutated FLT3 is constitutively active even in the absence of FLT3L. Constitutive activation causes perpetual activation of downstream signaling through the JAK/STAT5/PIM-1, RAS/MEK/MAPK/ERK, and PI3K/Akt/mTOR pathways, causing unchecked survival and growth of AML blasts. The purple and blue circles represent the Extracellular domain of the FLT3 receptor. The yellow circle with a P inside represents a phosphate. Arrows indicate the downstream pathways that are activated. The figure was generated using Biorender (https://www.biorender.com/).

**Figure 2 ijms-25-09448-f002:**
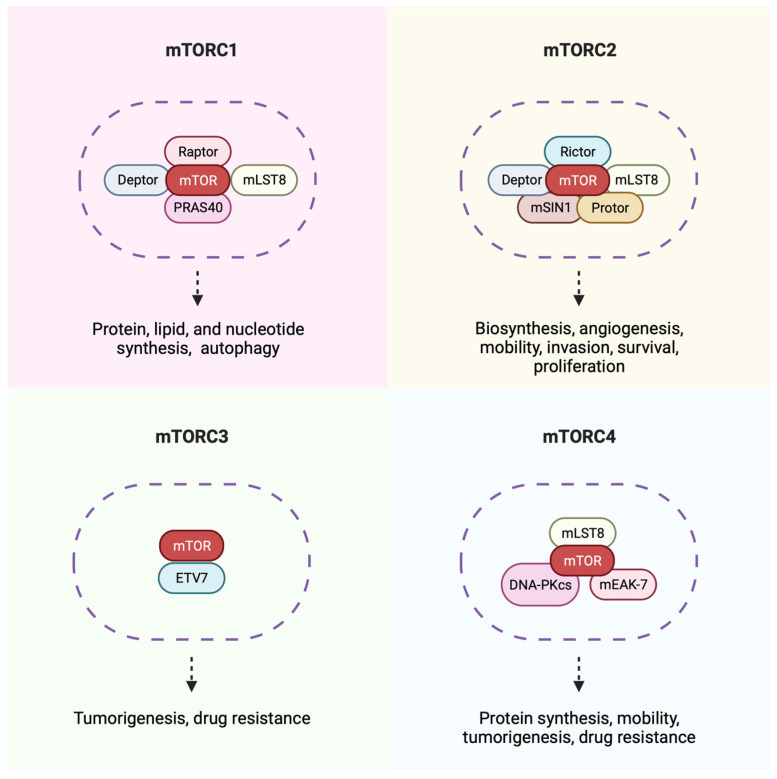
Overview of the four complexes found in the mTOR pathway. Specific components of each complex and their potential physiological functions are denoted in the figure [97]. While all complexes contain mTOR as part of the central structure, they differ in their composition of activating proteins. mTORC1 contains Raptor, Deptor, mLST8, and PRAS40. mTORC2 contains Rictor, Deptor, mLST8, MSIN1, and Protor. Not much is known about mTORC3; however, it contains ETV7 in its structure. mTORC4, like mTORC1 and mTORC2, contains mLST8, but also mEAK-7 and DNA-PKcs as part of its structure. Unchecked signaling through these complexes causes tumorigenesis and unchecked survival and proliferation. The figure was generated using Biorender.

**Figure 4 ijms-25-09448-f004:**
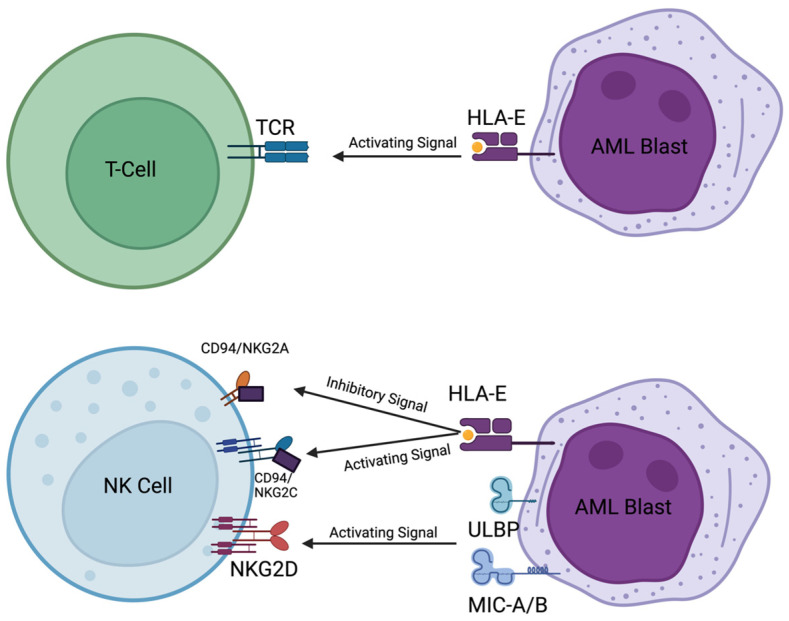
T cell versus NK cell engagement with AML blasts. The HLA-E peptide complex binds to inhibitory receptor NKG2A/CD94 or activating receptor NKG2C/CD94 on the NK surface. The activating receptor NKG2D binds to ULBPs and MICA/MICB. The figure was generated using Biorender.

**Table 1 ijms-25-09448-t001:** Overview of the FLT3 tyrosine kinase inhibitor family [58,70,71,72,73]. OS: overall survival; RFS: relapse-free survival; CIR: cumulative incidence of relapse; NRM: non-relapse mortality; GVHD: graft-versus-host disease; AE: adverse event.

Inhibitor	Type	Targets	Adverse	Outcome
First Generation				
Midostaurin	Type I	Other kinases (KIT, PDGFR, RAS/RAF/MEK, JAK)	Cytopenias, flu-like symptoms, cardiac failure (<5%), nausea, vomiting, diarrhea	OS, RFS
Sorafenib	Type II	Other kinases (KIT, PDGFR, RAS/RAF/MEK, JAK)	Cytopenias, flu-like symptoms, hypertension, cardiac ischemia (<5%), diarrhea, rash, hand-foot	OS, RFS, CIR, NRM, GVHD, AE
Second Generation				
Gilteritinib	Type I	Higher potency, more specific, less off-target	Cytopenias, diarrhea, pancreatitis, headaches	OS, RFS, CIR, AE
Quizartinib	Type II	Higher potency, more specific, less off-targe	Cytopenias, flu-like symptoms, cardiac arrythmia, nausea, anorexia	OS, RFS, AE

**Table 2 ijms-25-09448-t002:** Summary of FLT3 inhibitor clinical trials.

	Trial	Developmental Phase
First Generation Inhibitors		
Midostaurin + 7+3 chemotherapy	RATIFY (NCT00651261)	Phase III
Midostaurin + Azacitidine	Internal Trial at MD Anderson Cancer Center	Phase I/II
±Midostaurin + Allo-HCT	RADIUS (NCT01883362)	Phase II
Sorafenib	SORMAIN (DRKS00000591)	Phase II
Sorafenib + 7+3 chemotherapy	SORAML (NCT00893373)	Phase II
Second Generation Inhibitors		
Quizartinib + Cytarabine/Daunorubicin	NCT 01390337	Phase I
Quizartinib + Allo-HCT	2689-CL-0011	Phase I
Quizartinib + Allo-HCT	NCT00462761	Phase I
Quizartinib + 7+3 chemotherapy	QuANTUM-First (NCT02668653)	Phase III
Gilteritinib or salvage chemotherapy	QuANTUM-First (NCT02668653)	Phase III

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
