# Peer review of "Enhancing Therapeutic Efficacy of FLT3 Inhibitors with Combination Therapy for Treatment of Acute Myeloid Leukemia"

_ijms, 2024, doi:10.3390/ijms25179448_

Round 1

Reviewer 1 Report

Comments and Suggestions for Authors

The review article titled ( Using Flt3 inhibitors in combination with small molecules inhibitors in treatment of AMl ) written by Leifheit et al is very well written and updated but I have some comments 

1. Regarding the introduction section section please add some information about the the aim of the review as it is written in a very general way without focusing on the aim of the review 

2. Using the extramedullary spread is more convenient with AML than metastasis 

3. The authors can add a section about LSCs intramedullary niche and it role in relapse.

4. Adding a table that summarizes the preclinical trial and clinical trial done on the combination therapy will be a great add and reference for the readers

Otherwise, the manuscript is great, the references are updated and the figures is very illustrative 

Author Response

REVIEWER 1 COMMENTS:

The review article titled (Using Flt3 inhibitors in combination with small molecules inhibitors in treatment of AML) written by Leifheit et al is very well written and updated but I have some comments 

Regarding the introduction section section please add some information about the aim of the review as it is written in a very general way without focusing on the aim of the review.

Response:

  • Line 59-62 now reads"AML is a multifaceted disease, and in this review, we focus on the treatment for FLT3+ AML patients with an emphasize on using FLT3 inhibitors in combination with other therapies. Therefore, approaching it from many angles will optimize and enhance the therapeutic benefits of TKI treatment."

Using the extramedullary spread is more convenient with AML than metastasis.

Response: 

  • Lines 76-83. Leukemic stem cells (LSCs) are a major source for refractory disease or relapse. LSCs are therapy- resistant cells with persistent self-renewal capacity that drive clonal outgrowth and are essential in the context of extramedullary spread [14–16]. As well as being therapy-resistant [14–16], they are also marked by the acquisition of pro-survival and pro-proliferative mutations that may emerge as the dominant subclone throughout disease progression [17] and serve as a reservoir for disease relapse after conventional treatments [16,18].

The authors can add a section about LSCs intramedullary niche and it role in relapse.

Response: Added a section about LSCs intramedullarly niche and it role in relapse-Lines 93-105.

LSCs are a highly heterogenous population with a variety of mutations that contribute to their unregulated growth in the BM [21]. The bone marrow niche, regulated by crosstalk with a variety of cell types, functions to maintain stem cells and serve as a reservoir for hematopoiesis. However, LSCs also rely on this crosstalk within the BM niche to maintain their survival and proliferative properties [20–22]. A. variety of signaling pathways, including SDF1⍺/CXCR4, Wnt/β-catenin, VCAM/VLA-4/NFκB, and PI3K/Akt/mTOR pathways, positively support development of LSCs and contribute to drug and chemotherapy resistance [23]. AML relapse occurs due to therapy-resistant minimal residual disease (MRD), in which LSCs remain in the BM environment post-treatment [20,24]. These remaining LSCs contribute to relapse by maintaining their gene expression and properties that allow for stemness and self-renewal [25]. Therefore, identifying and targeting the LSC compartment is of utmost importance to prevent AML relapse following treatment.

Adding a table that summarizes the preclinical trial and clinical trial done on the combination therapy will be a great add and reference for the readers

Response:

Added a new tables that summarizes the preclinical and clinical trials  which appears as Table 2 in revised manuscript.

Otherwise, the manuscript is great, the references are updated and the figures is very illustrative 

Reviewer 2 Report

Comments and Suggestions for Authors

The manuscript is a review article focusing on AML, a type of blood cancer. The manuscript is dealing with various treatments of AML. It is very well written. However, there are some minor comments:

1. Line 106. What is FMS, can it be spelled when introduced first? Is it e.g. "Feline McDonough Sarcoma"? Overall, there are many abbreviations in the manuscript. Kindly make sure that all of them are spelled at least once or when needed (abstract, main text, Figure legend). An independent section with Abbreviations can be also considered.

2. Line 108. What is Runt? Can it be spelled?

3. The same for FLT(3)

4. Line 143. Figure legend for Figure 1. Kindly provide more details in the Figure legend which otherwise seems to be too concise. It will improve the clarity and accessibility of the article.

5. Table 1 is presented as a Figure. It is more beneficial to indeed make it as a text-based table, because it would allow the readers to click and access the references directly and work with text rather than image. 

6. Consider elaborating the text in Figure legend to Figure 2. 

7. Section 9 looks like one large paragraph. Consider dividing it into smaller paragraphs to simplify reading (9. Engaging Immune Cells with IL-15).

8. Usage of in vivo and in vitro. It is not consistent in the text, and italics are recommended for Latin phrases. 

9. Lines 560-561. What is the relevance of "animal protocols" to this review article? Unnecessary standard sections from the template, including this one, could be removed. 

10. "Conflict of interest" statement needs to be elaborated according to the journal's/MDPI standards. 

Comments on the Quality of English Language

Some abbreviations need to be spelled, usually only once in the Abstract, once in the main text, and once for the Figure legends. Currently, some abbreviations are spelled more than once, and some are not spelled.

Usage of "in vivo" and "in vitro" phrases is not consistent. Italics are expected. 

Author Response

REVIEWER 2

The manuscript is a review article focusing on AML, a type of blood cancer. The manuscript is dealing with various treatments of AML. It is very well written. However, there are some minor comments:

1. Line 106. What is FMS, can it be spelled when introduced first? Is it e.g. "Feline McDonough Sarcoma"? Overall, there are many abbreviations in the manuscript. Kindly make sure that all of them are spelled at least once or when needed (abstract, main text, Figure legend). An independent section with Abbreviations can be also considered.

Response: The sentence has been revised "The FMS-like tyrosine kinase 3 also known as FLT3 mutation affects one third of patients with AML and is correlated with poor prognosis."

2. Line 108. What is Runt? Can it be spelled?

Response: The RUNX1 gene provides instructions for making a protein called runt-related transcription factor 1 (RUNX1). Runt 

3. The same for FLT(3)

Response: this has been added - fms-like tyrosine kinase 3 (FLT3)

4. Line 143. Figure legend for Figure 1. Kindly provide more details in the Figure legend which otherwise seems to be too concise. It will improve the clarity and accessibility of the article.

Response: Figure 1. Structure of the FLT3 receptor in the inactive and active/mutated forms and associated downstream. FLT3 ligand (FLT3L) binding to the FLT3 ligand causes receptor dimerization and activation of downstream signaling pathways. Mutated FLT3 is constitutively active even in the absence of FLT3L. Constitutive activation causes perpetual activation of downstream signaling through the JAK/STAT5/PIM-1, RAS/MEK/MAPK/Erk, and PI3K/Akt/mTOR pathways, causing unchecked survival and growth of AML blasts. Figure created with Biorender.5. Table 1 is presented as a Figure. It is more beneficial to indeed make it as a text-based table, because it would allow the readers to click and access the references directly and work with text rather than image. 

6. Consider elaborating the text in Figure legend to Figure 2. 
Response: Figure 2. Overview of the four complexes found in the mTOR pathway. Specific components of each complex and their potential physiological functions are denoted in the figure.While all complexes contain mTOR as part of the central structure, they differ in their composition of activating proteins. mTORC1 contains Raptor, Deptor, mLST8, and PRAS40. mTORC2 contains Rictor, Deptor, mLST8, MSIN1, and Protor. Not much is known about mTORC3, however, it does contain ETV7 in its structure. mTORC4, like mTORC1 and mTORC2, contains mLST8, but also mEAK-7 and DNA-PKcs as part of its structure. Unchecked signaling through these complexes causes tumorigenesis and unchecked survival and proliferation. Figure created with Biorender.

7. Section 9 looks like one large paragraph. Consider dividing it into smaller paragraphs to simplify reading (9. Engaging Immune Cells with IL-15).

Response: Divided into two paragraphs

8. Usage of in vivo and in vitro. It is not consistent in the text, and italics are recommended for Latin phrases. 

Response: These have been changed to italics

9. Lines 560-561. What is the relevance of "animal protocols" to this review article? Unnecessary standard sections from the template, including this one, could be removed.

Response: this has been removed.

Reviewer 3 Report

Comments and Suggestions for Authors

In this manuscript entitled “Using FLT3 Inhibitors in Combination with Small Molecule Inhibitors for the Treatment of Acute Myeloid Leukemia”, the authors covered a broad range of topics in FLT3+ AML. However, a broad, general topic makes it difficult to talk about in depth. Given the fact of quite a few papers published on FLT3 AML recently, narrowing down the topic is needed. The following are my specific and major concerns.

1.     Regarding the Title: The title of the manuscript sounds very confusing to me because those FLT3 inhibitors are small molecule inhibitors, and some of the FLT3 inhibitor combinations/regimens the authors talked about, such as recombinant IL15, are NOT small molecule inhibitors. Therefore, the title is incorrect and needs to be changed.

2.     Regarding the Introduction: The introduction does NOT talk about FLT3 AML treatment at all. It does not mention anything about FLT3 inhibitor combination therapy in FLT3+ AML, which is the topic of the manuscript. It should be rewritten to provide a relevant introduction.

3.     Lines 116-118: The authors said “The FLT3 mutation has two forms: the FLT3 internal tandem duplication (FLT3-ITD) mutation and the FLT3 tyrosine kinase domain (FLT3-TKD) mutation. FLT3-TKD accounts for 25% of AML cases, while FLT3-ITD is seen in 5% of cases [33,34]”. I don’t think that the latter sentence is correct. As far as I know, FLT3-TKD is less common than FLT3-ITD. The correct statement should be: Mutations of FLT3 are found in approximately 30% of newly diagnosed AML cases and occur as either ITDs (≈ 25%) or point mutations in the TKD (7–10%). See Daver, et al. Targeting FLT3 mutations in AML: review of current knowledge and evidence. Leukemia 33, 299–312 (2019).

4.     Regarding the Table 1: The authors need to spell out those acronyms in the table of contents in the table legend. Some of the fonts are too small to read and try to use the same size font in the table text.

5.     The authors used quite a few very long sentences, for example, lines 31-34 “Acute myeloid leukemia (AML) is a heterogenous malignancy of the myeloid lineage stem cell precursors that give rise to red blood cells, platelets, white blood cells (not including B and T cells) and is characterized by chromosomal abnormalities, recurrent gene mutations, and epigenetic modifications, affecting chromatin structure[1,2].”

Lines 48-51 “A major source for refractory disease or relapse after patients reach remission are therapy-resistant cells with persistent self-renewal capacity that drive clonal outgrowth, known as (6) leukemic stem cells (LSCs) and are essential in the context of metastasis [7–9]”. Those long sentences are difficult to follow and understand. Please rephase the long sentences in the manuscript.

6.      What kind of the reference format was used? I have not reviewed any manuscripts using such a strange format without the volume and the pager number of a cited paper (see below.)

References

1. Pelcovits, A.; Niroula, R. Acute Myeloid Leukemia: A Review; 2020

Comments on the Quality of English Language

Please see my comments above.

Round 2

Reviewer 3 Report

Comments and Suggestions for Authors

The revised manuscript reads better than the previous one. The following are my concerns/comments.

1.     Regarding the Abstract: I would suggest rephasing the 1st sentences-“The Relapsed and refractory fms-like tyrosine kinase 3 (FLT3) + acute myeloid leukemia (AML) represents a challenging clinical scenario characterized by a high rate of relapse, even after allogeneic hematopoietic stem cell transplantation (allo-HSCT) to provide the definition of FLT3+ AML, i.e., FLT3-mutated. Of note, FLT3+ AML is commonly used in clinical conversations, but uncommonly used in publications.

2.     Regarding the Introduction: Lines 49-50 “The FMS-like tyrosine kinase 3 also known as FLT3 mutation…”, which is incorrect. It should be corrected to “The mutations in the gene encoding FMS-like tyrosine kinase 3 (FLT3)…”. Lines 50-52: “…increased proliferation and decreased apoptosis (reference)”. The reference is missing here.

3.     Lines 151 “…and tumor protein p53 (tp53)”. It should be corrected to TP53. Please go through the entire manuscript and check all the terms and abbreviations carefully.

4.     Regarding the Table 1: The fonts for “Adverse Effects” are too small to read, and the same size font should be used in the table text.

5.     Regarding the figure 3, it contains too many small sub-figures and is too complicated, and the fonts are too small. I am unable to follow the sub-figures and the text. The figure 3 needs to be modified.

Comments on the Quality of English Language

The English is fine.

Author Response

  1. Regarding the Abstract: I would suggest rephasing the 1stsentences-“The Relapsed and refractory fms-like tyrosine kinase 3 (FLT3) + acute myeloid leukemia (AML) represents a challenging clinical scenario characterized by a high rate of relapse, even after allogeneic hematopoietic stem cell transplantation (allo-HSCT) to provide the definition of FLT3+ AML, i.e., FLT3-mutated. Of note, FLT3+ AML is commonly used in clinical conversations, but uncommonly used in publications.

 Response: The first two sentences have been changed to read: FMS-like tyrosine kinase 3 (FLT3) mutations are genetic changes that are found in approximately thrity percent of patients with acute myeloid leukemia (AML). FLT3 mutations in AML represents a challenging clinical scenario characterized by a high rate of relapse, even after allogeneic hematopoietic stem cell transplantation (allo-HSCT)". These are in track changes in the revised manuscript.

  1. Regarding the Introduction: Lines 49-50 “The FMS-like tyrosine kinase 3 also known as FLT3 mutation…”, which is incorrect. It should be corrected to “The mutations in the gene encoding FMS-like tyrosine kinase 3 (FLT3)”. Lines 50-52: “…increased proliferation and decreased apoptosis (reference)”. The reference is missing here. Response: Lines 49-50 now read " The mutations in the gene encoding FMS-like tyrosine kinase 3 also known as FLT3 mutation affects one third of patients with AML and is correlated with poor prognosis (Ref 6)". The reference has been added to the text.
  2. Lines 151 “…and tumor protein p53 (tp53)”. It should be corrected to TP53. Please go through the entire manuscript and check all the terms and abbreviations carefully. Response: this has been corrected and the manuscript has been checked for other abbreviations.
  3. Regarding the Table 1: The fonts for “Adverse Effects” are too small to read, and the same size font should be used in the table text. Response: The font size has been increased.
  4. Regarding the figure 3, it contains too many small sub-figures and is too complicated, and the fonts are too small. I am unable to follow the sub-figures and the text. The figure 3 needs to be modified. Response: Figure 3 has been modified into figure 3A and figure 3B and the font size has been increased.

Round 3

Reviewer 3 Report

Comments and Suggestions for Authors

See the attached comments.

Comments on the Quality of English Language

There are a few errors.

Author Response

Reviewers comments: This revised manurcipt reads better than the previous one. However, there are still some errors. Please go through the manuscripts carefully. For examples, the comma in the first senetnce in the abstract should be removed. See below. “Abstract: FMSfms-like tyrosine kinase 3 (FLT3) mutations, are genetic changes that are found in 14 approximately thirty percent of patients with + acute myeloid leukemia (AML). FLT3 mutations in 15 AML represents…” Figure 3 is still problematic. Response: We have gone through the manuscript and corrected various errors highlighted in the edited version with track changes.

Reviewers comment: I don’t understand why the authors included three sub-figures in Figure 3A. They look redundant to me. See below: Response: The original figure 3 was divided into A). Pro-survival pathways in AML and B). Pro-survival pathways inhibited by FLT3i and mTORi.

There is only Figure 3 A and B. In the edited version it may look like there is another figure but in the unmarked version (pdf) there is only two parts to figure 3. This figure was divided into A and B to make it easier to read.
